# Impact of Bullying—Victimization and Gender over Psychological Distress, Suicidal Ideation, and Family Functioning of Mexican Adolescents

**DOI:** 10.3390/children9050747

**Published:** 2022-05-19

**Authors:** Silvana Mabel Nuñez-Fadda, Remberto Castro-Castañeda, Esperanza Vargas-Jiménez, Gonzalo Musitu-Ochoa, Juan Evaristo Callejas-Jerónimo

**Affiliations:** 1Department of Psychology, Coast University Center, University of Guadalajara, Puerto Vallarta 48280, Mexico; remberto.castro@academicos.udg.mx (R.C.-C.); esperanza.vargas@academicos.udg.mx (E.V.-J.); 2Department of Education and Social Psychology, University of Pablo de Olavide, 41013 Seville, Spain; gmusoch@upo.es (G.M.-O.); jevaristocallejas@gmail.com (J.E.C.-J.)

**Keywords:** bullying victimization, gender, psychological distress, suicidal ideation, family functioning, adolescents

## Abstract

Bullying victimization is strongly associated with increased psychological distress and suicide in adolescents and poor family functioning. Knowledge of gender differences influencing these factors will improve the prevention of mental problems and suicide in victimized adolescents. A total of 1685 Mexican secondary students, 12–17 years old (*m* = 13.65), of whom 54% were girls, responded to a standardized scale questionnaire to analyze such differences. Based on the statistical analysis, girls reported significantly lower family functioning and higher psychological distress and suicidal ideation than boys. The cluster analysis classified adolescents into high (5.78%), moderate (24.07%), and no-victimization (69.76%) groups. Boys predominated in the high (3.1%) and moderate-victimization (12.4%) clusters, and girls in the no-victimization group (39.51%). Multivariate statistical analyses found significant differences between the three groups, with the highest means of psychological distress and suicidal ideation and lowest family functioning in the high-victimization group. Only for suicidal ideation, there was an interaction between gender and the degree of victimization, with girls showing a higher increase of suicidal ideation than boys in the same cluster. Conclusions: Early detection and intervention in bullying-victimized adolescents, aiming to decrease psychological distress and suicidal ideation and strengthen family functioning, should consider contextual gender differences for effective prevention of mental health problems and suicide in adolescents.

## 1. Introduction

Bullying is a type of violence against pairs characterized as being unprovoked, repetitive over time, and in the context of a power imbalance favoring the aggressor. Bullying victimization refers to the situation in which a student receives such acts of violence from one or several of their fellows (the bully/bullies) repeatedly over time. Usually, the perpetrators choose students at a disadvantage, finding it difficult or impossible to overcome the situation [1]. Adolescents can be involved in bullying as perpetrators (bullies), victims, bully victims (victims that also bully others), or witnesses that can either support the bully, defend the victim, or are unresponsive [2]. Bully behavior can be physical (toward the victims and their belongings), verbal, and relational (spreading rumors, excluding from social activities and groups). The increased use of electronic devices in young people has opened access to harassment through virtual social nets, known as cyberbullying [3].

The bioecological model of human development [4] helps organize and understand reciprocal interactions and changes of multiple factors involved in bullying at the individual, relational (family, school, community), structural, and cultural–historic levels. This theoretical model proposes that development depends on reciprocal interactions between individuals (ontosystem), their family, school, and community (microsystem) and between these environments (mesosystem), influenced by structural (exosystem) and socio-cultural factors (macrosystem), changing over time (chronosystem). Gender, the social construction of identity assigned to men and women, determines differences in involvement and the effects of bullying in adolescents and the multiple associated factors. The influence of gender on adolescent psychosocial adjustment and mental health should also be examined in the frame of their diverse cultural contexts, aiming to enhance our knowledge to tailor contextual-gender prevention policies and programs [5].

The global prevalence of bullying victimization is 35.3%, with wide variations in prevalence and gender differences [6] according to regions and countries. Most current research and knowledge about bullying comes from developed countries, and developing countries are underrepresented [6,7]. In more patriarchal and traditional cultural contexts, as in east Asia and Latin America, boys are more involved in bullying than girls as perpetrators and victims in any modality [8,9,10]. Some studies indicated that girls tend to use more relational forms and boys more physical and verbal violence, according to differences in gender expression of negative emotions. Recent findings point out that such differences are blurred in particular socio-cultural contexts [11] and are evolving according to historical changes in awareness of gender inequities [7].

Bullying is harmful to all the participants, including the witnesses, but affects the victims more. Its adverse effects on adolescent development and health in the short and long term were confirmed consistently by research, the most worrying being the impact on mental health and its relation to adolescent suicide [12,13,14]. Research have highlighted that these negative effects are related to the frequency and overall duration of bully victimization, differentiating high or severe victimization (always or several times in a period) from moderate victimization (few times or sometimes).

Previous studies found that victimization significantly impacts adolescents’ mental health [15]. That includes depressive and anxiety disorders [10,16] and externalizing problems, such as substance use, violent and antisocial behaviors, and sexually risky behaviors [17,18]. An efficient measure to evaluate general mental health is psychological distress (PD), defined as the suffering present when the stress overcomes personal coping resources [19]. It manifests as depressive and anxious symptoms, ranging from mild to severe. Its spectrum includes subclinical problems, umbral cases, and mental disorders [20,21]. Women and girls have worse mental health, reporting higher means of psychological distress, depression, and anxiety disorders than men and boys [5], a situation that is referred to as the “gender mental health gap.”

Bullying victimization increases the risk of suicidal behavior, including adolescent suicidal deaths [6,12]. Adolescent suicide is currently the fourth leading cause of death in adolescents and represents a quarter of overall annual deaths by suicide [22,23]. Suicidal ideation accounts for a 12.1–33% lifetime prevalence, and non-fatal suicide attempts occur in 4.1–9.3% of adolescents based on international estimations [24]. Suicidal ideation (SI) is the cognitive component of suicide behaviors that goes from thinking of death as a solution to the idea of taking the own life and planning how to do it (suicide planning). Knowing the intermediate links that lead from suicide ideation to suicide attempts is critical in evaluating suicidal risk [13,25]. Some degree of suicidal ideation is common in early adolescence. If it is brief and infrequent, it does not represent a risk factor for suicide, but when frequent and persistent, it is predictive of subsequent suicidal attempts in a one-year follow-up period [25,26]. Gender has a particular role in determining the risk for suicidal behaviors. While women have twice or more SI, suicide planning, and attempts than men, men die by suicide two to five times more than women in any age group [27].

Considering that half of the adolescents’ deaths due to suicide occurs in the first attempt, effective suicidal prevention should point at detection in the earlier stages and lower risk than suicide attempts [26]. Associated factors that facilitate the transition from ideation to action are violence victimization (by family violence or bullying) [13,15], impulsivity, and conditions that can increase it (alcohol or drug use) [12,28], mental disorders [13,24], lack of social support, and loneliness [6,29,30,31]. Hong et al. [29] proposed five factors that can explain the trajectory of bullying to suicide: depression, anxiety, low self-esteem, loneliness, and hopelessness. On the other hand, social connectedness and support, especially from the family, are the most protective factors against adolescent suicide [6,30,31,32].

In adolescence, family relationships remain the main provider of social support, influencing bullying involvement [33,34], psychological distress, and suicidal behavior. Good family functioning (FF), the perception of open communication, parental warmth, support, and acceptance have strong effects against bullying involvement, directly and through its negative relationship with psychological distress [35,36,37]. It also moderates the impact of school victimization and other stressful situations on adolescents [38,39] and is a main protective factor for adolescent suicide [13,32,40]. On the contrary, lack of support, harsh parental practices, offensive communication, and family violence will have a direct effect on increasing the involvement in bullying as an aggressor, victim, or bully victim [35,37,41] and also increase the risk for suicidal behavior through emotional dysregulation, depression, anger, and social skills deficits [42].

FF has to adjust in response to multiple changes that require mutual adjustment between parents and children during adolescence. The perception of FF varies according to age, with a worse perception in early adolescents due to increased conflicts and the search for more autonomy [43]. It can differ in girls and boys, depending on contextual gender stereotypes that set different rules, responsibilities, and expectations for them from their mothers and fathers [44,45]. Girls tend to report a worse perception of family functioning than boys and more conflicts with their mothers [44]. The effect of family functioning on bullying victimization is well established, but we did not find research about how school victimization could affect adolescent family functioning.

In this study, we aim to identify the influence and interaction between different degrees of bullying victimization and gender over perceived FF, PD, and SI in early adolescence in the context of the Mexican culture. Considering the current knowledge summarized, we formulated our hypothesis. 

**Hypothesis** **1** **(H1).***There will be significant differences in PD, SI, and perception of FF between boys and girls*.

**Hypothesis** **2** **(H2).***There will be statistically significant differences in adolescents’ psychological distress, suicidal ideation, and family functioning perception according to the frequency of victimization*.

**Hypothesis** **3** **(H3).***There will be an interaction effect between victimization and gender, making the effects on PD, SI, and FF worse for girls than for boys in the same victimization group*.

## 2. Materials and Methods

It is a quantitative, cross-sectional, observational study. To obtain baseline measurements of psychosocial adjustment and social support of high school students in our context, a large exploratory study was designed based on Bronfenbrenner’s ecological model. The project included multiple psychosocial adjustment and maladjustment variables and family, school, and community variables. A single measurement was made, which characterizes it as cross-sectional because we wanted to know the variables with the greatest weight for adjustment of the adolescents and their interactions. The data presented in this work belong to this larger project.

### 2.1. Participants

Over the total of secondary school students (*n* = 14,759) in Puerto Vallarta (México) for a 2.5% error, with a 0.5 variance and a 95% confidence level, the representative sample was 1685 students. The ages of the participants were between 12 and 17 years old (M = 13.65, DT = 1.14), and 54% were girls. We randomly selected the participating schools from the official list provided by education local authorities, using a by-staged conglomerates-sampling method [46]. The participating centers were ten public and three private schools.

### 2.2. Instruments

The school victimization scale [47] has 22 Likert-type items, with three sub-scales: verbal, physical, and relational victimization. Items 21 and 22 ask about the frequency of victimization in one year. The factorial analysis confirmed a three-dimensional structure, with good adjustment to the data (SBχ^2^ = 293.7139, gl = 142, *p* < 0.001, CFI = 0.952, RMSEA = 0.031 (0.026, 0.036)). The Cronbach’s alpha reliability in this study was excellent with 0.95 for the full scale, 0.92 for relational victimization, 0.72 for physical victimization, and 0.85 for verbal victimization.

The Psychological Distress Scale K10 of Kessler [21], Spanish version [20], composed of ten items, with one to five Likert options, ranging from never to always, measures the presence of depression and anxiety in the previous week. The factorial analysis found a unidimensional structure and adequate feet to the data (SBχ^2^ = 120.9903, gl = 30, *p* < 0.001, CFI = 0.983, RMSEA = 0.043 (0.045, 0.051)), and excellent reliability (0.95 by Cronbach’s alpha).

Robert’s Suicidal Ideation Scale, Spanish adapted version, was validated on the Mexican population [48]. The scale has four items with four options according to the frequency of ideation from 0 (not a day) to 4 (five to seven days) over a week. The Cronbach’s alpha was 0.85 in this study.

Apgar family functioning [49], Spanish version [50], includes five items examining adolescents’ perceptions of their family adaptation, participation, growth, affection, and problem resolution. For this study, the exploratory and confirmatory factor analyses showed a unidimensional structure that reflects adolescent satisfaction with family functioning and good adjustment to data (SBχ^2^ = 6.0817, gl = 3, *p* < 0.05, CFI = 0.998, RMSEA = 0.025 (0.000, 0.053)). Reliability by Cronbach’s alpha was 0.85.

### 2.3. Procedure

The research followed the bioethical requirements of the Helsinki Declaration [51]. The research team contacted the students’ tutors, sending them a letter, explaining the purpose and procedure of the research and asking for their consent. The letter included a format for the tutor’s signature, authorizing the adolescent to participate. The scales were applied by the research team in March and April 2017 in the students’ classrooms over a one-hour period. Researchers explained the study’s goals to the students and voluntary and anonymous participation. Some authorized students (1%) refuse to participate. Additionally, eleven questionnaires were excluded due to missing (8) or atypical values (4), resulting in 1687 subjects for the final sample. If the missing values accounted for less than 20% of the scale, we applied the regression imputation procedure [52] but excluded the scale if they are higher than 20%. Subjects with more than two excluded scales were omitted from this study. We followed standardized scores and Mahalanobis distance estimations to detect atypical data [53].

### 2.4. Data Analysis

In the descriptive statistical analysis, the means and standard deviations of the variables studied had significant differences by gender, confirmed by the *t* Student proof. The Pearson analysis found significant correlations between all the variables. Bi-stage cluster analysis grouped the students into three clusters according to the frequency of bullying victimization: high victimization (HV), moderate victimization (MV), and no victimization (NV). A multivariate analysis of the variance (MANOVA) with victimization and gender as independent variables predicted differences in the adolescents’ PD, SI, and family functioning. Finally, we used the Bonferroni test to determine minimal significative distances between means of victimization-gender combinations. Statistical analyses were performed with SPSS 25 package.

## 3. Results

### 3.1. Correlations

Pearson correlations were statistically significant at a level of 0.01 for all the variables in the study (Table 1). Correlations between FF, PD, and SI were relevant and negative with *r* coefficients −0.396 and −0.389. Victimization factors correlated positively with PD and SI, with the higher *r* values for relational victimization (0.377 for PD and 0.316 with SI). Verbal victimization correlated higher than physical with PD (0.365) and SI (0.315).

### 3.2. Cluster Analyses

Bi-stage cluster analysis classified adolescents into three groups: high victimization (HV) with 97 adolescents (5.78%), moderate victimization (MV) with 404 adolescents (24.07%), and no victimization (NV) with 1177 adolescents (69.76%). As Table 2 shows, the no-victimization cluster has a higher percentage of girls, and boys predominate in the moderate- and high-victimization groups.

### 3.3. MANOVAS

A 2 × 3 MANOVA was performed with bullying victimization and gender as independent variables and PD, SI, and family functioning as the dependent variables. The main effects and interactions reached statistical significance (Table 3).

#### 3.3.1. Main effect: victimization 

Differences between victimization clusters (Table 4) were significant for all the dependent variables, showing that no-victimized adolescents reported the lowest means of PD and SI. The group of moderate victimization obtained higher means for PD and SI than the no-victimized one but lower than the high-victimization cluster that showed the highest means for PD and SI. The perception of FF also presented significant differences among the three clusters, with the lowest mean of FF in the high-victimization group and the higher perception of FF for the no-victimization cluster.

#### 3.3.2. Main effect: gender

Regarding the differences by gender (Table 5), girls reported significantly lower means than boys in perceived FF and higher means than boys in PD and SI.

#### 3.3.3. Interaction

There was an interaction between victimization and gender for SI with girls displaying higher increases of SI than boys in each victimization group (Figure 1).

The post-hoc Bonferroni test confirmed that girls score significantly higher on SI in every cluster. Furthermore, the mean of SI for girls in the group of moderate victimization was higher than boys’ SI in the high-victimization group (Figure 2). 

## 4. Discussion

Our results showed that Mexican boys are more victimized than girls in accordance with gender differences found in other countries [7,8,10]. The percentage of adolescents victimized was 29.85%, lower than the global estimation of 35.3% reported in a recent survey [6]. Interestingly, correlations confirmed that in early adolescence, social exclusion has even more impact on mental health than verbal and physical victimization [14,54,55] and highlighted the protective effect of satisfactory family functioning over PD and SI.

Mexican girls reported higher means for PD and SI than boys and lower satisfaction with family functioning, as we stated in our first hypothesis (H1), consistent with the worldwide reports of the mental health gender gap [5,6]. The gender gap is currently associated with an increase in social pressure, expectations, and stress between women and girls. However, gender bias in the report of perception, identification, and communication of emotions can contribute to the disparity of these results. As in other patriarchal socio-cultural contexts, Mexican men are supposed to be stoic, and the expression of feelings, such as sadness and loneliness, could be criticized and labeled as feminine. That can also explain the fact that because girls acknowledge and communicate better their PD and SI, it could be easier for them to ask and seek help from their relational support systems as well at the mental health system. Receiving help could protect girls against consummated suicide better than boys [56,57]. Another gender-related difference is that boys tend to respond to violence with externalizing behavior problems more often than girls [25]; adding to suicidal risk is alcohol consumption, which facilitates impulsive attempts, even in the absence of frequent SI [28,58]. In the Mexican cultural context, the early onset of alcohol use for boys is extended and socially accepted. However, recent surveys showed an alarming increase in adolescent girls’ alcohol use, surpassing boys (26.6% vs. 22.5%), while boys’ excessive consumption remains higher than girls’ (22.3% vs. 14.7%) [59]. Awareness about gender inequities keeps growing and expanding globally, changing the perception and behavior of girls and boys; for instance, recent research on emotional expression in high-stress contexts shows that girls can react equally with boys in verbal and physical expressions of anger, instead of sadness [11]. 

The fact that girls expressed more dissatisfaction with FF is critical for their worse mental health indicators. In the Mexican context, mothers currently have more access to employment. However, they are still the main providers of care (for children and older people) and domestic work [60], issues that have not yet been solved, even in more gender-equal contexts [5,61]. Consequently, mothers are more central for Mexican adolescents’ wellbeing and can replicate gender inequities, demanding domestic support to daughters better than sons and being more restrictive with girls’ permissions. That leads to increased conflicts and resentment between mothers and daughters [62]. Latino girls are expected to be more responsible and respectful of family values and rules than boys [5,63]. As their mothers, girls are overloaded with pressure to accomplish independence and productive participation, in addition to traditional gender expectations [5].

Statistically significant differences in PD, SI, and FF between the three victimization clusters confirmed our second hypothesis (H2). The lowest mean for PD and SI corresponded to the no-victimized group. In contrast, adolescents in the moderate-victimization cluster reach higher self-reported PD and SI levels than the no-victimized but lower than the highly victimized adolescents that reported the maximum levels of PD and SI. These results added to previous findings, confirming that bullying victimization impacts adolescents’ mental health proportionally to the frequency of bullying [6,13,15,64]. Additionally, it allows for predicting that Mexican adolescents with high victimization will have more PD and SI than those who are moderately victimized or not victimized. Perception of FF followed an inverse pattern, with the highest values for the no-victimized students, decreasing in the moderate-victimization cluster and reaching the lowest satisfaction with FF in the high-victimization cluster. This particularity is crucial in Mexican adolescents given the importance of family support, suggesting that bullying victimization could increase the suicide vulnerability by two complementary paths: increasing PD and decreasing the protective power of family relationships. Further studies can address the possibility of bidirectional effects, helping to design more comprehensive interventions, considering that FF can buffer the impact of victimization on PD and SI. In addition, lowering PD would help to improve FF in early adolescence, amplifying the beneficial interactions between adolescents and their parents.

Our third hypothesis (H3) was partially confirmed. As our results show, bullying victimization will interact with gender, making the effect on suicidal ideation remarkably higher in girls than in boys. Girls with moderate victimization reached higher means of SI than boys with high victimization. The interaction was not present for PD and FF, meaning that girls and boys respond equally with more PD and worse FF when bullied. Bullying victimization, especially the relational type, leads to a low-relational evaluation that would be experienced as psychological pain correlated with self-injurious and suicidal thoughts and behaviors [55]. Additionally, dissatisfaction with family function, the most potent protective factor against stress in our context [65], could explain the accentuated impact of psychological pain in girls’ suicidal ideation.

Considering these differences in light of mental health gender knowledge, the lower level of SI in victimized boys could not ascertain lower suicidal risk. At least in our socio-cultural context, psychological distress, accompanied by a careful evaluation of other main facilitators of suicide attempts, such as alcohol use and impulsive conduct, could be more robust indicators to evaluate suicidal risk in victimized boys. Given that the antibullying programs are generally school based, a regular evaluation of PD, SI, and FF could help identify and guard adolescents with suicidal risk. Strategies for increasing adolescent mental wellbeing should be ecological, pointing to the quality of microsystems. That means eradicating relational violence inside the family, school, and community and structural violence and inequities determined by gender, diversities, race, religion, poverty, etc. These factors could overlap, potentiating the damage to adolescents’ mental health.

As violence is one of the main factors related to mental health problems, efforts should be directed to foster the learning of nonviolent ways to relate with each other inside families, schools, and communities. Relational forms of bullying can be more challenging to tackle, considering that they are less evident than verbal or physical types. Although teachers can find it unreasonable to try to influence the relational preferences and friendship connections between their students, strengthening social inclusion should not be resigned. Additionally, it is critical that antibullying programs include screening of adolescents’ PD, SI, and FF, to bring support for those affected and their families. It will help to identify those who needs further evaluation, especially regarding suicide risk and specialized care when needed.

The present study has some limitations to consider. It included only early adolescents and results could vary in middle or late adolescence, according to changes in family functioning and psychological distress. The rapid physical and psychosocial changes linked to puberty results in a marked increase in PD [5] and lowered the satisfaction with FF in early adolescence [43], mainly because of increased conflicts in family relationships. In most adolescents, these changes attenuate in middle and late adolescence because family relationships successfully change and stabilize over time, and learning effective coping skills help to decreases PD. The Mexican socio-cultural context will determine differences in these results compared with other countries, but it is also a strength, given the scarce information about bullying victimization and gender influence on adolescent mental health in Latin America. For suicidal behavior, only suicidal ideation was evaluated and not suicidal planning or attempts nor other risk indicators, such as self-injury or alcohol use. Finally, these results should be enhanced by a structural model and tested by longitudinal designs to confirm causality. Our research contribution to theory confirms the relation of the degree of victimization over early adolescents’ mental health and perception of FF and the gender-determined differences in the intensity of SI, contrasting with PD that has no significant differences between boys and girls. This suggests that for girls SI scores can be an adequate estimation of suicide risk, while for boys it will be more accurate to identify those with high PD and low FF to further explore associations with other risk factors, such as impulsive behavior and alcohol consumption. These results can inform the design and evaluation of prevention and gender-sensitive early intervention and programs, contributing to applied research on bullying and adolescent mental health studies.

## 5. Conclusions

The results of this research highlight the differential impact that the degree of victimization and gender has on Mexican adolescents’ psychological distress, suicidal ideation, and perceived family functioning. While boys are more victimized, girls in every cluster (no victimization, moderate victimization, and high victimization) reported higher means of PD and SI and lower FF than boys. The interaction between gender and degree of victimization was confirmed for suicidal ideation as higher for girls than for boys. Research suggests that boys can be under-reporting SI and PD because of gender stereotypes on the perception and communication of emotions, such as sadness and fear, that can be labeled as feminine. Adolescent girls tend to report worse perceived family functioning than boys, which could be related to higher expectations, responsibilities, and restrictions assigned to girls in Mexican culture, increasing suicidal ideation. Accordingly, it is important to consider gender cultural differences in the evaluation and interpretation of data regarding PD, SI, and FF to design contextualized strategies that can effectively reach adolescents and their families to improve the prevention of mental problems related to bullying victimization, especially adolescent suicide.

## Figures and Tables

**Figure 1 children-09-00747-f001:**
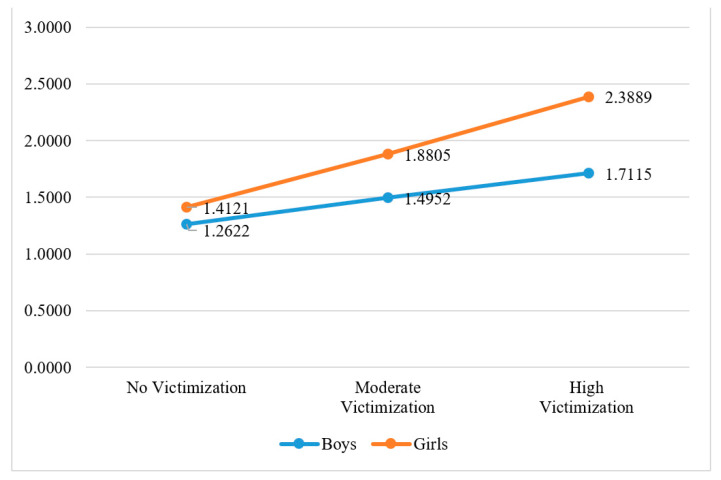
Marginal means of suicidal ideation and gender.

**Figure 2 children-09-00747-f002:**
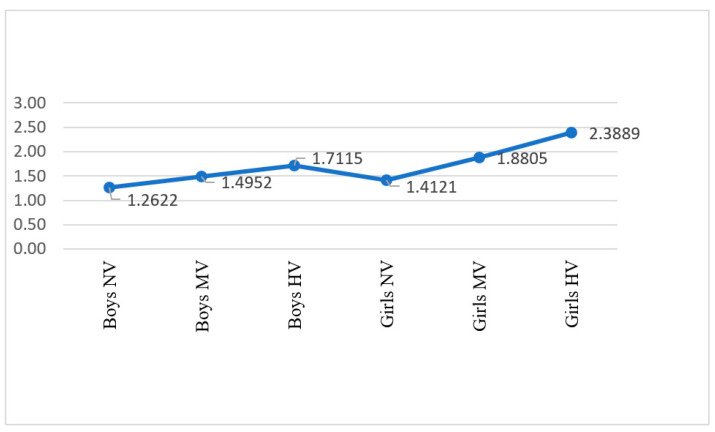
Marginal means of suicidal ideation and gender. Note: NV: no victimization, MV: moderate victimization, HV: high victimization.

**Table 1 children-09-00747-t001:** Pearson correlations, means, and standard deviations.

	1	2	3	4	5	6
1. Relational Victimization	1					
2. Direct Physical Victimization	0.709 **	1				
3. Direct Verbal Victimization	0.838 **	0.720 **	1			
4. Psychological Distress	0.377 **	0.244 **	0.365 **	1		
5. Suicidal Ideation	0.316 **	0.236 **	0.315 **	0.534 **	1	
6. Family Functioning	−0.186 **	−0.168 **	−0.184 **	−0.396 **	−0.389 **	1
Means	1.66	1.46	1.73	2.29	1.47	3.61
Standard Deviation	0.78	0.63	0.78	0.84	0.71	1.09

** Correlation is significant to 0.01 level (bidirectional).

**Table 2 children-09-00747-t002:** Cluster Distribution by Victimization and Gender.

Gender	HV (N y%)	MV (N y%)	NV (N y%)	Totals by Gender
Boys	52 (3.1%)	208 (12.4%)	514 (30.63%)	774 (45.88%)
Girls	45 (2.68%)	196 (11.68%)	663 (39.51%)	904 (53.58%)
Total	97 (5.78%)	404 (24.07%)	1177 (69.76%)	1678 (100%)

**Table 3 children-09-00747-t003:** MANOVAS by victimization and gender.

Variation Source	Variables					
	Λ A	F	df_entre_	df_error_	*p*	η^2^
(A) Victimization ^a1,a2,a3^	0.858	44.453	6	3340	<0.001 ***	0.074
(B) Gender ^b1,b2^	0.952	27.796	6	1670	<0.001 ***	0.059
A × B	0.987	3.760	6	3340	<0.001 ***	0.007

Note: ^a1^: no victimization, ^a2^: moderate victimization, ^a3^: high victimization; ^b1^: boy, ^b2^: girl. *** *p* < 0.001.

**Table 4 children-09-00747-t004:** ANOVAS in function of victimization.

Source of Variation	Bullying Victimization				
	NV	MV	HV	F	η^2^
Psychological Distress	2.12 (0.77)	2.62 (0.85)	3.00 (0.89)	97.872 ***	0.104
Suicidal Ideation	1.35 (0.61)	1.68 (0.79)	2.02 (0.96)	70.125 ***	0.077
Family Functioning	3.73 (1.05)	3.40 (1.10)	3.13 (1.29)	24.179 ***	0.028

Note: *** *p*< 0.001; NV: no victimization; MV: moderate victimization; HV: high victimization.

**Table 5 children-09-00747-t005:** ANOVAS in function of gender.

Source of Variation	Gender			
	Boys	Girls	F	η^2^
Psychological Distress	2.10 (0.76)	2.46 (0.88)	76.250 ***	0.043
Suicidal Ideation	1.36 (0.58)	1.56 (0.78)	37.271 ***	0.813
Family Functioning	3.78 (0.99)	3.47 (1.15)	35.876 ***	0.021

Note: *** *p*< 0.001.

## Data Availability

The data presented in this study are available on request from the corresponding authors.

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
