# Peer review of "Impact of Bullying—Victimization and Gender over Psychological Distress, Suicidal Ideation, and Family Functioning of Mexican Adolescents"

_children, 2022, doi:10.3390/children9050747_

Round 1
Reviewer 1 Report
The article meets the requirements and is of high quality. It would be great if you could explain in detail in section 2 why you chose quantitative, cross-sectional, observational study.
Author Response
Response to Reviewer 1
We, the authors, acknowledge and appreciate the reviewer’s suggestions, which helped us improve our paper's quality.
Reviewer 1 said:
“It would be great if you could explain in detail in section 2 why you chose quantitative, cross-sectional, observational study.”
Response:
We included a paragraph in Section 2 (from 134 to 141) as follows:
“To obtain baseline measurements of psychosocial adjustment and social support of high school students in our context, a large exploratory study was designed based on Bronfenbrenner’s ecological model. The project included multiple psychosocial adjustment and maladjustment variables and family, school, and community variables. A single measurement was made, which characterizes it as cross-sectional because we wanted to know the variables with the greatest weight for adjustment of our adolescents and their interactions. The data presented in this work belong to this larger project”.
Reviewer 2 Report
Can authors clarify the differences in their study speaking to adolescent differences in reporting depending on their age. ?
“The present study has some limitations to take into account. It included only early 327 adolescents, and results could vary in middle or late adolescents, according to changes in 328 family functioning and psychological distress “
Can authors consider explaining what is “Bullying victimization “?
Can authors clarify what is “according to the level of victimization and the difference between high and moderate levels”
Author Response
Response to Reviewer 2
We, the authors, acknowledge and appreciate the reviewer’s suggestions, which helped us improve our paper's quality.
Reviewer 2 said:
1. Can authors clarify the differences in their study speaking to adolescent differences in reporting depending on their age?
“The present study has some limitations to take into account. It included only early 327 adolescents, and results could vary in middle or late adolescents, according to changes in 328 family functioning and psychological distress “.
Response:
We included an explanation as follows:
“The rapid physical and psychosocial changes linked to puberty resulted in a marked increase in PD [5] and lowered the satisfaction with FF in early adolescence [44], mainly because of increased conflicts in family relationships. In most adolescents, these changes attenuate in middle and late adolescence because family relationships successfully change and stabilize over time, and learning more effective coping skills helps to decrease PD. (From 339 to 334)
Reviewer 2 said:
“2. Can authors consider explaining what is “Bullying victimization “? “
Response:
We included the Bullying victimization definition in the introduction as follows:
“Bullying victimization refers to the situation in that a student receives such acts of violence from one or several of their fellows (the bully/ bullies) repeatedly over time. Usually, the perpetrators choose students at a disadvantage, that find difficult or impossible to overcome the situation [1]” (31 to 33)
Reviewer 2 said:
- Can authors clarify what is “according to the level of victimization and the difference between high and moderate levels”
Response:
We explained the levels by frequency of victimization in the Introduction as follows.
“Research has highlighted that these negative effects are related to the frequency and overall duration of bully-victimization, differentiating high or severe victimization (always or several times in a time) from average victimization (few times or sometimes). (From 66 to 69) Additionally, we changed the word “level” to “frequency,” which is more descriptive and precise (130).